# Introduction: Special Issue on the Visual International Relations Project

**Sarah Gansen * and Patrick James ***

Political Science and International Relations, University of Southern California, Los Angeles, CA 90089-0044, USA
* Correspondence: gansen@usc.edu (S.G.); patrickj@usc.edu (P.J.)

**Abstract:** The application of systemism, an innovative and user-friendly technique for generating lucid, graphic summaries of analytical arguments, can enhance the social sciences. Content, as research and pedagogy move forward, becomes increasingly vast and diverse in theory and methods. Systemism offers both a means and a method for enhanced communication in the face of challenges posed by the rapid expansion of the social sciences in the fast-paced world of the new millennium. This is the motivation for a Special Issue of *Social Sciences* that will show systemism in action. The Visual International Relations Project (VIRP) archive continues to accumulate materials. The contents of this Special Issue will demonstrate the value of that resource across a wide range of subject areas. This introductory article proceeds in five sections. The first section provides a general overview of systemism and the VIRP. The second section introduces systemism in greater detail as a graphic approach to the communication of ideas. The third section applies systemism to convey the framework for analysis utilizing a classic work of social science—*The Logic of Collective Action*. The fourth section outlines the articles that follow in making up the Special Issue. The fifth and final section sums up what has been accomplished in this introductory article.

**Keywords:** bricolagic bridging; collective action; systemism; systematic synthesis; Visual International Relations Project

## 1. Overview

The application of systemism, an innovative and user-friendly technique for generating lucid, graphic summaries of analytical arguments, can enhance the social sciences. Content, as research and pedagogy move forward, becomes increasingly vast and diverse in theory and methods. Systemism offers both a means and a method for enhanced communication in the face of challenges posed by the rapid expansion of the social sciences in the fast-paced world of the new millennium. This is the motivation for a Special Issue of *Social Sciences* that will show systemism in action.

Systemist graphics of over 800 books and articles appear in the online archive of the Visual International Relations Project (VIRP) (www.visualinternationalrelationsproject.com accessed on 24 August 2023). The figures that appear in the articles from this Special Issue are among those in the ever-expanding VIRP archive. The archive is inclusive in terms of theoretical orientation, subject matter, and methods. While primarily based on publications that have appeared in English, there is also a great interest in (and some effort already has been made for) the inclusion of scholarship in other languages. (One of the 13 visualizations used in this Special Issue is based on a publication in a language other than English). Authors represented in the archive also span the globe, and the pieces visualized cover many decades. Items in the VIRP archive depict books, articles, and book chapters that connect with a wide range of disciplines in the social sciences and even beyond in the humanities and natural sciences.

By providing greater accessibility for material from academic publications, the graphic technique seeks not only to reduce barriers to communication in words alone but also to

assist scholars from different regions around the world. These scholars ultimately might even benefit the most from the implementation of systemism. Thus, the VIRP not only promotes rigorous discussion on a broad range of subjects within IRs and even throughout the social sciences, the archive also provides a space and a point for scholars from all over the world to engage with each other and grow in the field, which could be specifically advantageous for underrepresented groups and scholars early on in their careers.

This introductory article proceeds in four additional sections. Section 2 introduces systemism in greater detail as a graphic approach to the communication of ideas. The third section applies systemism to convey the framework for analysis with a classic work of social science—*The Logic of Collective Action* (Olson 1965). Section 4 outlines the articles that follow in making up the Special Issue. The fifth and final section sums up what has been accomplished in this introductory article.

## 2. Systemism

What is systemism? It is a way of thinking about theory that emphasizes completeness and logical consistency, achieved via attention to all possibilities for cause and effect across levels of analysis (Bunge 1996). Rather than a substantive theory, systemism is an *approach* (Bunge 1996, p. 265). It focuses on building comprehensive explanations; systemism transcends individualism and holism as the other available 'coherent views' with respect to the operation of a social system (Bunge 1996, p. 241). The essence of systemism in recent application to the discipline of International Ielations (IRs) is its emphasis on *graphic* communication to obtain those goals.[1]

Systemism as a method emphasizes the diagrammatic exposition of cause and effect that promotes comprehension and rigor. Thus, the overall value of systemism is that its visual representations clarify relationships expressed in a theory. Systemism goes beyond holism and reductionism using a focus on *all* types of connections needed to fully specify a theory. Systemism thereby facilitates the comparison of alternative visions with a clear and comprehensive presentation that also features succinctness. Consequently, systemism is both an *approach* and a *method* that, through its lucidity and completeness, has the potential to benefit the discipline as a whole.[2] The value of systemism includes both research-related and pedagogical applications.

Figure 1 outlines functional relations in a social system from a systemist point of view.[3] The varying shapes and colors that appear in the figure will be explained later on.

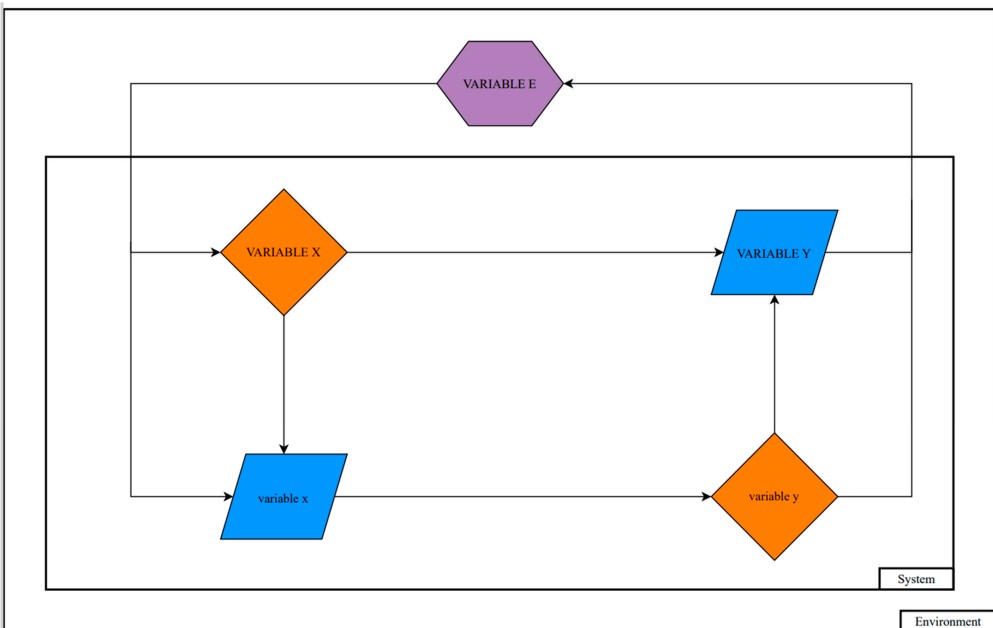

**Figure 1.** Functional relations in a social system. Source: Adapted from Bunge (1996, p. 149).

Figure 1 depicts the system and its environment. Variables that operate at macro (VARIABLE X, VARIABLE Y) and micro (variable x, variable y) levels of the system appear at its upper and lower levels.[4] In this diagram and others based on systemism, *UPPER- and lower-case characters correspond to MACRO- and micro-level variables*, respectively. The macro and micro levels are designated in line with the research question at hand. For example, consider a study of the foreign policy of Italy. A natural designation for the macro and micro levels would be the government and society of that State as a system. The environment for Italy would then be the international system beyond its borders.

Various other combinations are possible for system and environment, well beyond those such as state and international system that are found in the VIRP archive. Take, for instance, a study of Sociology as a discipline. The field of Sociology would become the system. The discipline as a whole, and individual scholars within it, would correspond, respectively, to the macro and micro levels. The world beyond Sociology would serve as its environment. In all instances, boundaries for system and environment should reflect what is anticipated to be most helpful in the visualization of the logic of a particular argument. This variation in the diagrams is welcomed to ensure the utmost level of lucidity and accuracy when producing an individual diagram while at the same time adhering to a clearly defined, overarching structure to facilitate the integration of paradigms and methods.

Four basic types of linkages are possible for macro and micro variables with each other: macro–macro (VARIABLE X → VARIABLE Y), macro–micro (VARIABLE X → variable x), micro–macro (variable y → VARIABLE Y), and micro–micro (variable x → variable y). The figure also includes a variable to represent the environment (VARIABLE E). The environment can be expected to stimulate the system and vice versa: (i) 'VARIABLE E → VARIABLE X' and 'VARIABLE E → variable x', and (ii) 'variable y → VARIABLE E' and 'VARIABLE Y → VARIABLE E'. All potential types of connection for a theory to incorporate are now in place.[5]

Table 1 provides the notation for systemist figures. Color and shape are used to designate roles for variables. The functions of variables as they appear along pathways are explained, wherever possible, using a river as an analogy.

**Table 1.** Systemist notation.

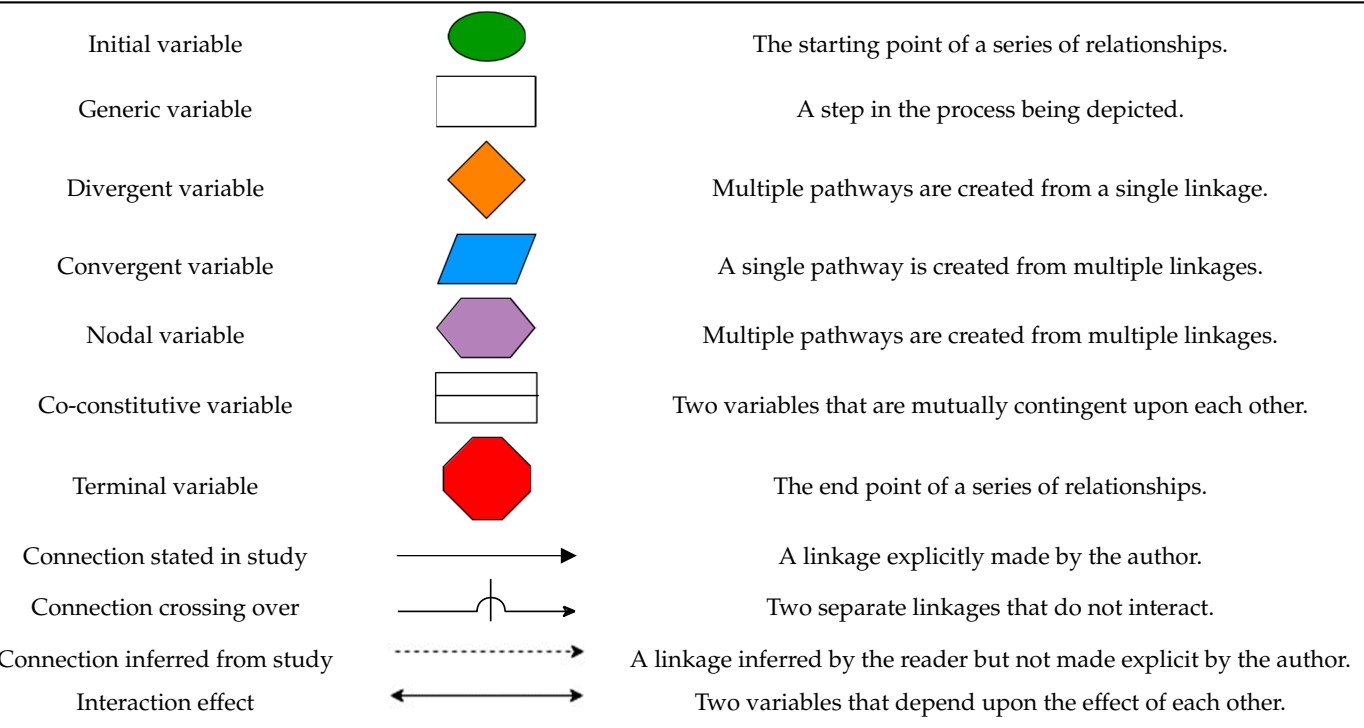

| | | |
|---|---|---|
| Initial variable | 🟢 | The starting point of a series of relationships. |
| Generic variable | ▭ | A step in the process being depicted. |
| Divergent variable | 🔶 | Multiple pathways are created from a single linkage. |
| Convergent variable | ▱ | A single pathway is created from multiple linkages. |
| Nodal variable | ⬡ | Multiple pathways are created from multiple linkages. |
| Co-constitutive variable | ▤ | Two variables that are mutually contingent upon each other. |
| Terminal variable | 🛑 | The end point of a series of relationships. |
| Connection stated in study | → | A linkage explicitly made by the author. |
| Connection crossing over | → | Two separate linkages that do not interact. |
| Connection inferred from study | ⇢ | A linkage inferred by the reader but not made explicit by the author. |
| Interaction effect | ↔ | Two variables that depend upon the effect of each other. |

An initial variable takes the form of a green oval, while a terminal variable is depicted as a red octagon. In virtually all instances, these designations are arbitrary and based upon starting and end points for argumentation within a given text. For the point of initiation, it always is possible to think of something else prior and, in principle, imagine effects beyond what is marked as the point of termination. The initial and terminal variables, respectively, are analogous to the source and mouth of a river.

With exactly one connection coming in and out, a generic variable appears as a plain rectangle. This might be thought of, in the context of a river, as a simple flowing forward—neither a point of convergence nor divergence for streams of water.

Three variable types—divergent, convergent, and nodal—include contingency. Represented with an orange diamond, a divergent variable signals the onset of multiple pathways. It is analogous to the emergence of multiple streams from a single river. Indicated with a blue parallelogram, a convergent variable designates a bringing together of multiple pathways. It is the same as a point of confluence for multiple rivers that then continue in a single stream. A purple hexagon denotes convergence followed by divergence of pathways—a nodal variable. In the riparian context, this would refer to multiple streams joining together and then coming apart. A co-constitutive variable—one with mutually contingent variables—appears in bifurcated form. This type of variable is included to recognize co-constitution as an essential aspect of social constructivist theorizing in particular and normatively oriented scholarship in general.[6]

Line segments are depicted in different ways, depending on what they are supposed to represent, and are explained as relevant within the respective figures. All of this combines to facilitate the creation of graphic summaries with high mutual commensurability and thus the potential to assist in managing a field's complexity and attendant communication-related challenges.

Return for a moment now to Figure 1, and the notation within it becomes fully intelligible. There are divergent and convergent variables—orange diamonds and blue parallelograms, respectively—at both the macro and micro levels of the system. The lone variable in the environment is of the nodal variety—a purple hexagon.

With notation in place, several observations can be made at this point about the relevance of systemist graphics to cause and effect. Systemism, in a word, is unforgiving. Its graphic format requires arguments to be made explicitly, with errors of either omission or commission being much easier to detect than from words alone. The structure of an analysis comes into bold relief. Another way of thinking that can be facilitated by systemist visualization is counterfactual analysis. Those arguments that obviously concern events that only take place in the abstract and can be especially challenging to understand are in a particular position to benefit from a parallel graphic exposition.[7]

With so many analytical frameworks and methods already available, it might be reasonable to ask this question: Who needs systemism? Once answered, the question itself seems ironic; it is precisely the scope and scale of research in place that makes the adoption of an approach such as systemism a major priority for the social sciences in the new millennium.

Systemist graphics can be implemented in three basic ways. One application is to a specific publication, namely, to tell its story about cause and effect in graphic form. This can begin with a pencil sketch and move through stages that include consultation with (one of) the author(s) of the respective piece.[8] The process culminates in a diagram that conveys the arguments of a given publication. The elaboration of such a diagram can take place by deleting, inserting, or revising existing connections. In carrying out such work, a broken-line notation is used, as per the listing in Table 1. Two other techniques also exist within systemism: systematic synthesis and bricolagic bridging. Each is introduced briefly in turn.

Systematic synthesis refers to the assembly of causal connections from a set of studies into one representative diagram. Thus, systematic synthesis is in line with the logic of confirmation—identifying the degree of consensus that exists about a given research question. One example would be the systematic synthesis of studies from the field of education that focus on a flipped classroom (Gansen and James 2022b). A flipped classroom refers to one that reverses activities normally occurring inside and outside of the scheduled meeting times

for a course. A basic version includes partially or even fully recorded lectures that students see prior to class. Within class time itself, discussions and various forms of active learning take place instead of a structured presentation from the instructor. Based on a sample of eight studies of the flipped classroom, Gansen and James (2022b) conducted a systematic synthesis that culminated in a visualization that combined arguments from across the sample of studies. A graphic of this kind can be subsequently accessed as a type of visual 'literature review'. It could also be used to summarize an area of the academic literature within a lecture or discussion section by a professor or teaching assistant. Furthermore, a systematic synthesis might be of value as a graphic overview and aid to memory—to either an undergraduate student studying for a test or a doctoral student preparing for qualifying exams. Systematic synthesis could be carried out for a topic in *any* social science in order to advance communication.

Bricolagic bridging might be introduced most easily with an antinomy—explaining its meaning using a point of contrast. In architecture, a plan is put forward; materials are obtained; and some kind of structure is assembled. The end product is expected to correspond as clearly as possible to what appears in the design. Bricolage, by contrast, consists of assembling materials in the absence of a plan. Thus, it is in line with the logic of discovery—generating new ideas rather than testing hypotheses. The priority in bricolagic bridging is to take studies that normally would not be engaged with each other and see if assembly into a single diagrammatic exposition stimulates new ways of thinking. By implementing this approach, respective social sciences could counteract tendencies toward segmentation and even isolation within a vast and expanding field of study.

One example of bricolagic bridging in International Rrelations involves engaging a prominent study of conflict processes with the literature on realist theories of war (James 2022, pp. 558–73). A review of *A Study of Crisis* (Brecher and Wilkenfeld [1997] 2000), a classic work on international conflict, produced valuable ideas for moving forward the realist research agenda. For example, the defensive variant of realism lacks any connection purely at the micro level. *A Study of Crisis* includes a wide range of micro-level variables, such as perception of stress for decision makers (James 2022, p. 570). Bricolagic bridging in James (2022) includes a plan for how defensive realism could achieve advancement with the incorporation of micro-level connections that access perception-based variables from *A Study of Crisis*. Much more could be said about the bricolagic bridging carried out in James (2022), which also includes ideas about how realist theories can stimulate research on conflict that goes beyond *A Study of Crisis*, but that would be beyond the scope of the present exposition.

### 3. Envisioning a Classic: The Logic of Collective Action

Among the all-time most impactful works of social science is *The Logic of Collective Action* (Olson 1965).[9] With origins in Economics, the theory put forward in Olson (1965) has exerted sustained influence throughout the social sciences. Against basic intuition about the effect of group size, the book argues that majority interests suffer because of the superior ability of minorities to coordinate and overcome the problem of collective action. The basic problem facing large groups such as voters or consumers, as opposed to business cartels or unions, is that of free riding. Based on relatively accessible equations and diagrams, Olson (1965) makes a convincing case that the interests of large groups often remain latent because individuals have a basic incentive to avoid making the effort to provide a collective good. The publicness of the good is the source of the problem—once created, it will be enjoyed by a group member whether or not that particular person made any effort toward its provision. Individuals, by contrast, must pay directly in order to obtain private goods, and that is the most basic contrast. In sum, smaller groups are advantaged over larger ones with regard to pursuing and obtaining collective goods.

Figure 2a a systemist graphic, tells the story of cause and effect from Olson (1965). The state is designated as the system, with the international system as its environment. Government and society correspond to the macro and micro levels, respectively. Note the presence of lower- and upper-case characters for the micro- and macro-level variables in the state as a system. The network includes nine micro and two macro variables in the state, with one variable in the international system.

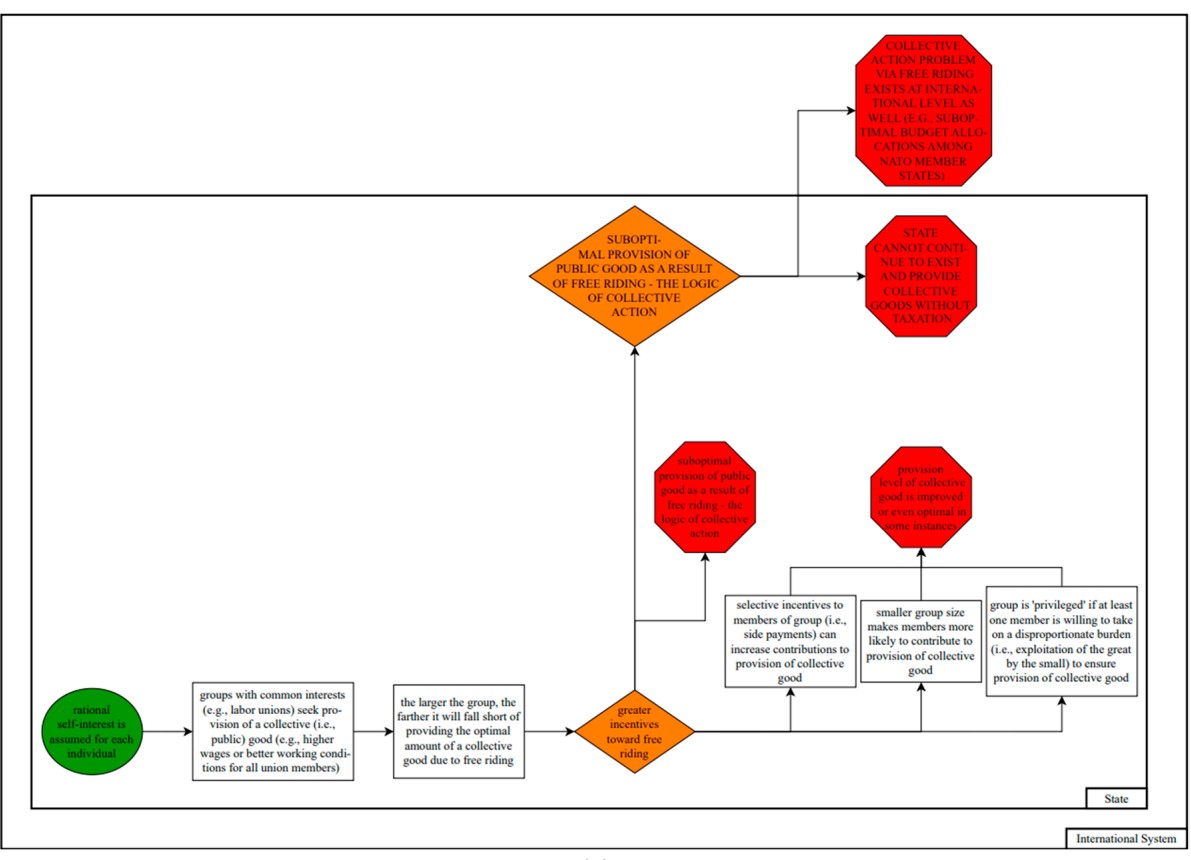

(**a**)

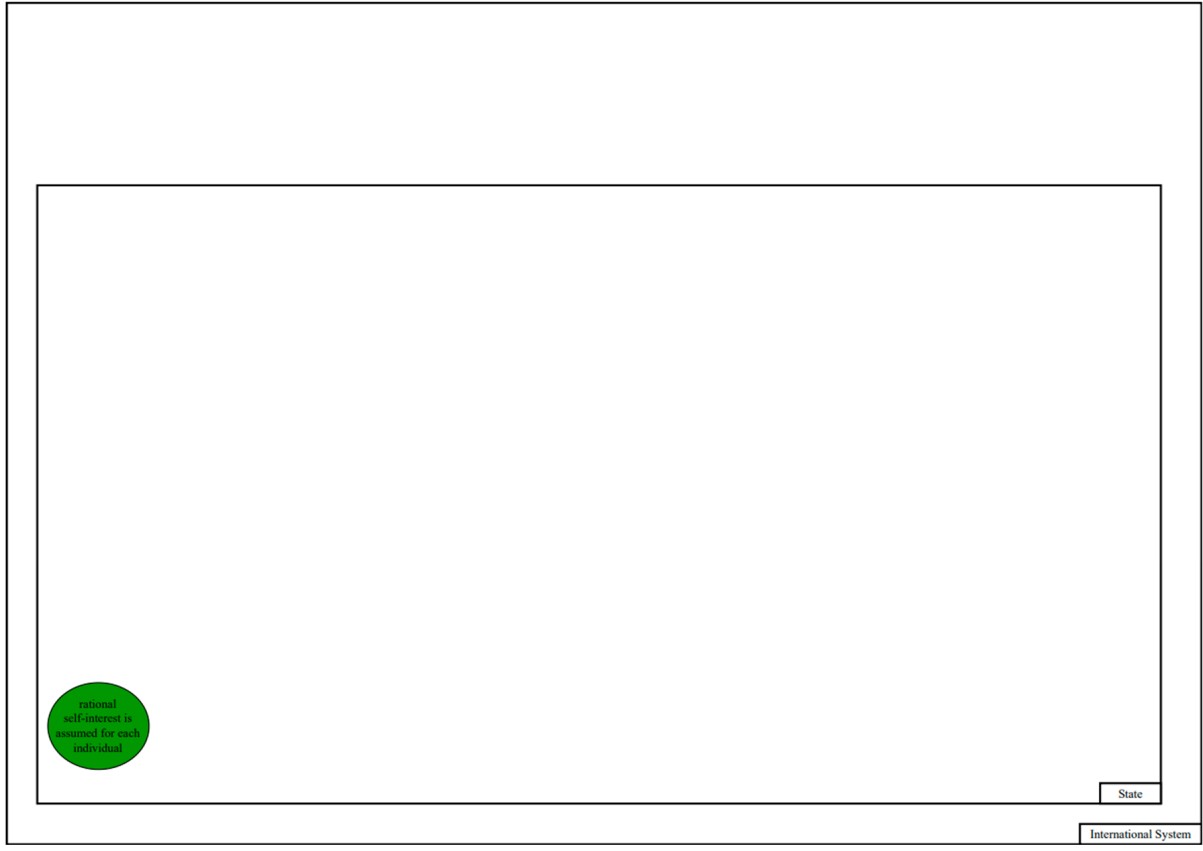

(**b**)

**Figure 2.** *Cont.*

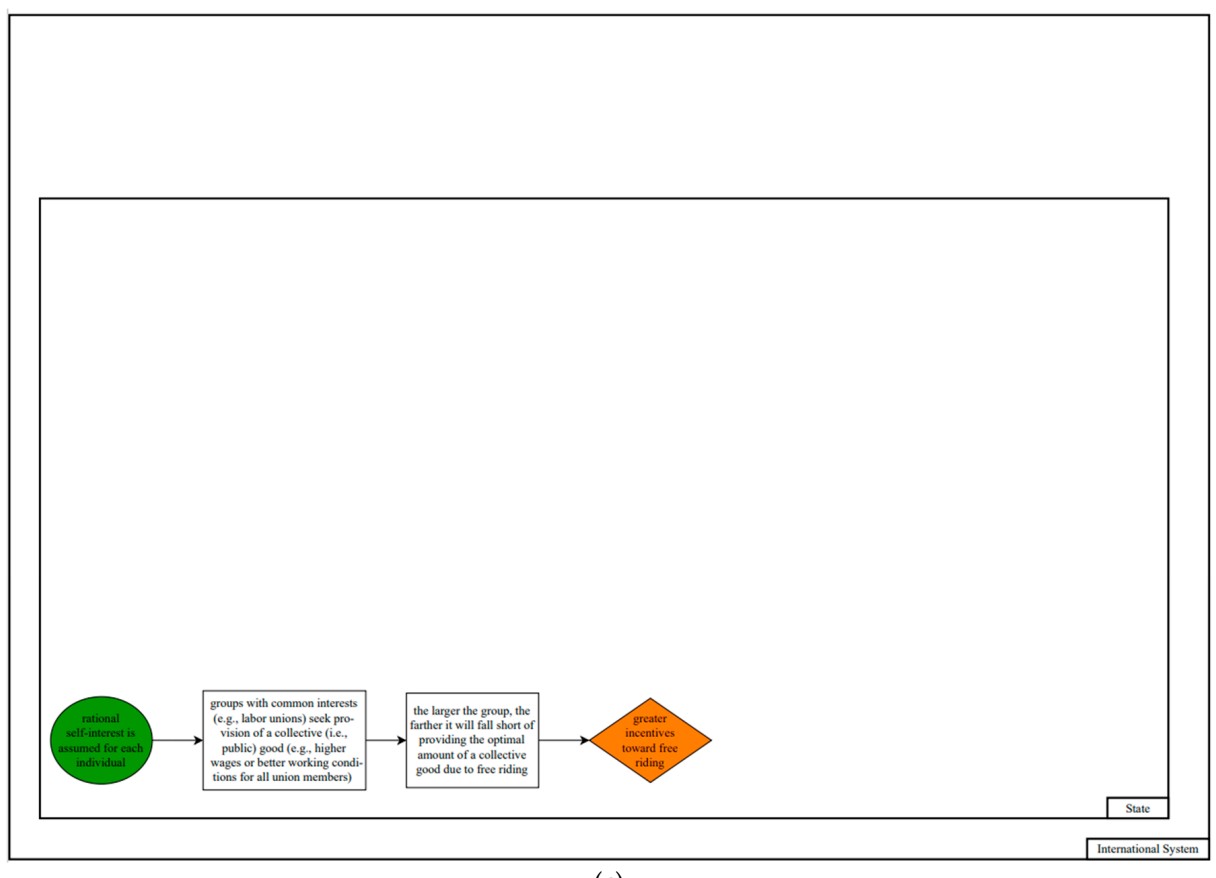

(**c**)

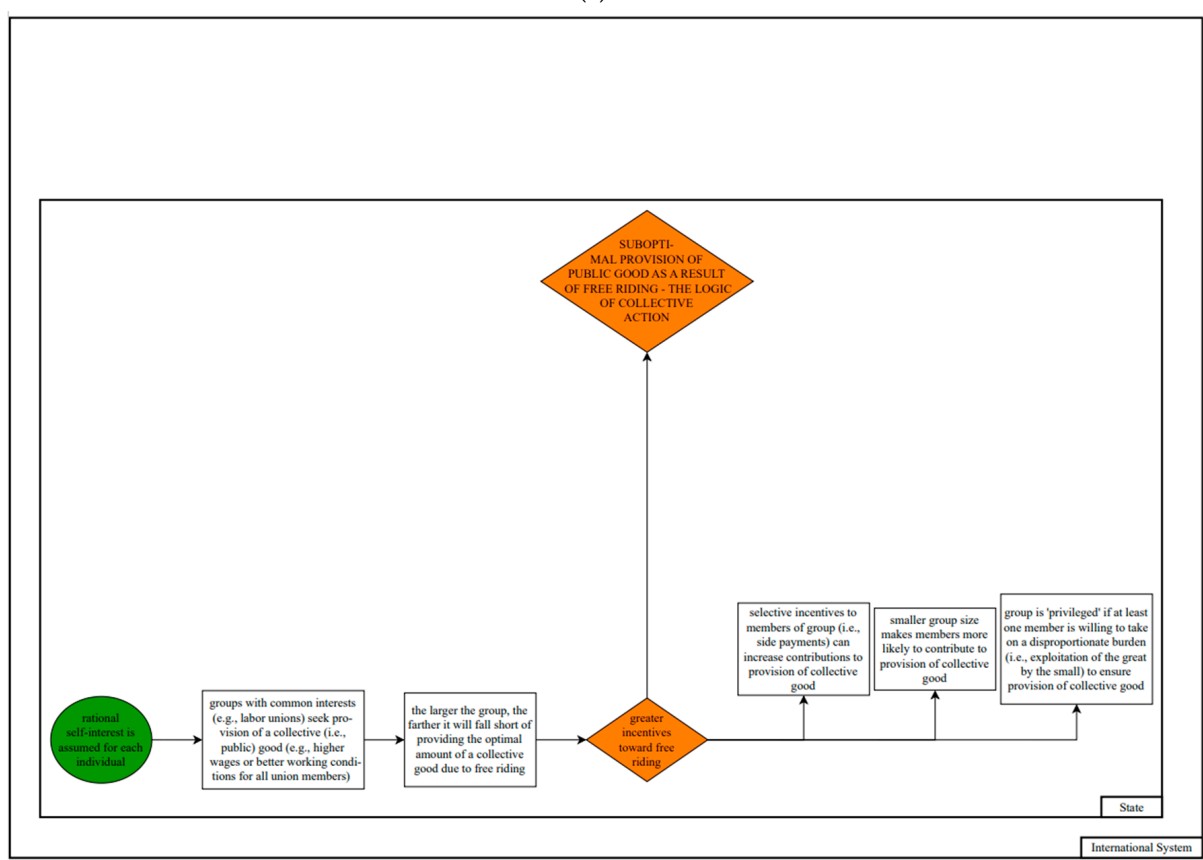

(**d**)

**Figure 2.** *Cont*.

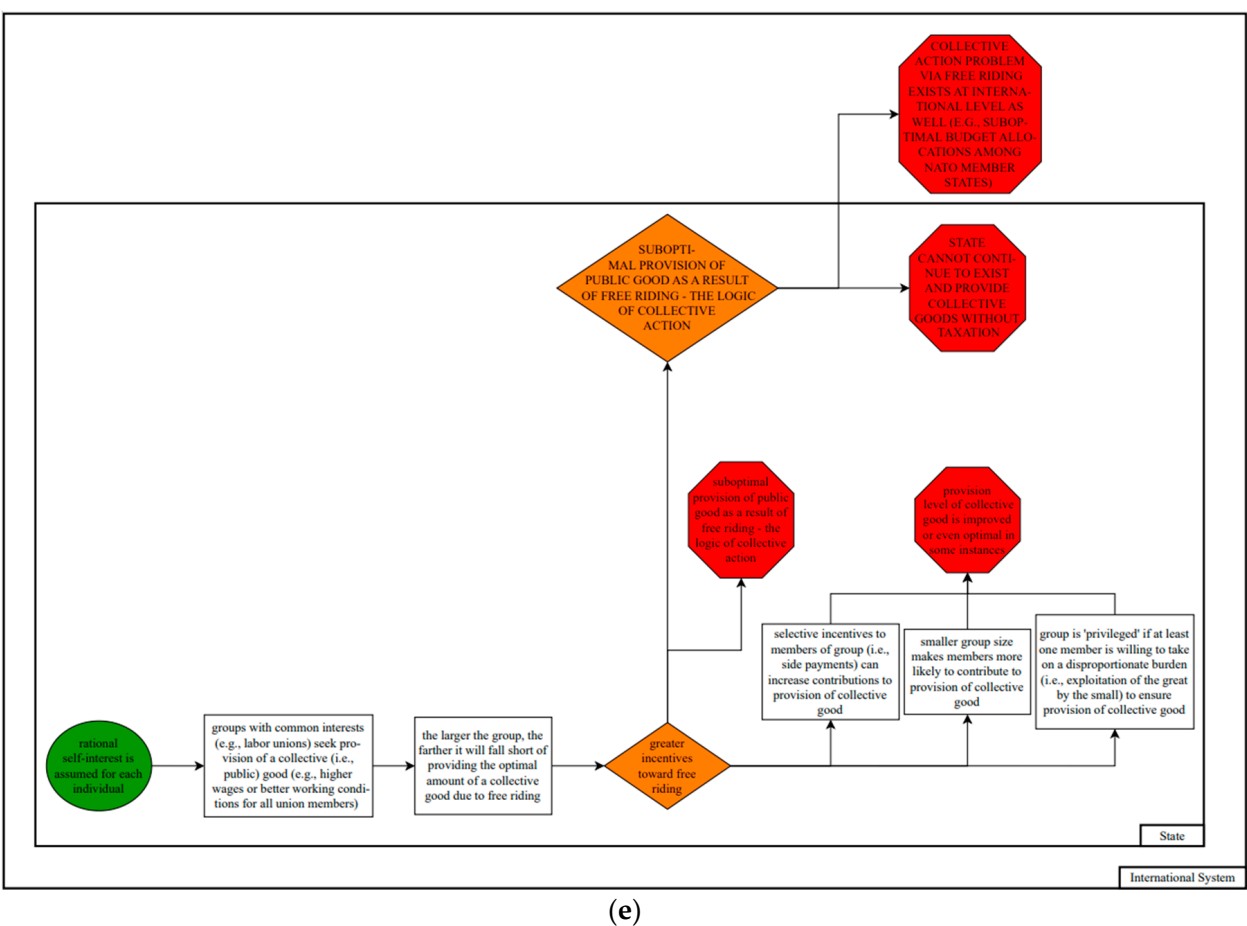

(**e**)

**Figure 2.** (**a**–**e**) The logic of collective action: public goods and theory of groups (Olson 1965). Diagrammed by: Sarah Gansen and Patrick James.

Among respective types of connection for a systemist diagram, four appear in Figure 2a. Note that all of the arrows are horizontal or pointing upward. This is because the original formulation of the theory of collective action focused primarily on processes within a given state. Arrows pointing downward—theorizing in line with Gourevitch (1978), for example, who focused on the impact of forces in the international system on domestic politics—had yet to develop in the research enterprise on collective action at the point of introduction in Olson (1965).

Figure 2a is presented in stages, with Figure 2b–e. Before moving through those sub-figures, a natural question arises about the layout. What are the criteria for placing one or more connections into a given sub-figure? The answer is purely pragmatic—ease of explanation. Some linkages or variables make more sense to put forward alone, while others may be explained with greater effectiveness in combination with each other. This is not an exact science, of course, and in principle, it is possible to move through a series of connections in different ways that produce the same overall result.

Figure 2b begins the assembly of sub-graphics into the overall story of cause of effect in Figure 2a. The state is the system, and the international system is the environment, in Figure 2b. This figure also includes an initial variable at the micro level that appears as a green oval: 'rational self-interest is assumed for each individual'. The assumption of rational choice is at the bedrock of Economics as a discipline, so this is a natural point of departure for the analysis.

Figure 2c shows the initiation of a micro-level pathway: 'rational self-interest is assumed for each individual' → 'groups with common interests (e.g., labor unions) seek provision of a collective (i.e., public) good (e.g., higher wages or better working conditions for all union members)'. This connection explains why only a 'closed shop' union, in

which membership is mandatory, has the potential to bargain effectively with management. Workers otherwise would have the incentive to free-ride and obtain benefits such as higher wages without paying union dues. The pathway continues with 'the larger the group, the farther it will fall short of providing the optimal amount of a collective good due to free riding'. The ultimate example, perhaps, is the Tragedy of the Commons (Hardin 1968). Humanity, collectively speaking, is responsible for preserving the environment. The tragedy referred to by Hardin (1968), put simply, is that when everyone is responsible for something in principle, then no one is in practice. When literally billions of people are involved, damage to the environment can be expected to continue because the incentive to free-ride is at its effective maximum for each person.

Figure 2c reaches a point of divergence, 'greater incentives toward free riding', which appears as an orange diamond. One branch continues on in Figure 2d at the micro level, with three connections: 'selective incentives to members of group (i.e., side payments) can increase contributions to provision of collective good'; 'smaller group size makes members more likely to contribute to provision of collective good'; 'group is 'privileged' if at least one member is willing to take on a disproportionate burden (i.e., exploitation of the great by the small) to ensure provision of collective good'. Each of the three preceding variables represents a mechanism for overcoming the free-rider problem. For example, in the case of a privileged group, consider the fans in a city who want to obtain an expansion team for one of the major sports. These franchises cost billions of dollars, so when successful expansion occurs, there normally is an ownership team consisting of a small number of extremely rich people who put up the money. Everyone in the city then benefits from watching the team play once it is up and running.

Another branch in Figure 2d moves up to the macro level: 'greater incentives toward free riding' → 'SUBOPTIMAL PROVISION OF PUBLIC GOOD AS A RESULT OF FREE RIDING—THE LOGIC OF COLLECTIVE ACTION'. Members of society per se are not expected to voluntarily contribute to supporting the activities of the national government. Without mandatory taxation, free riding would be the default position among individual citizens.

Figure 2e shows how each pathway reaches its respective point of conclusion. The points of conclusion appear as red octagons, indicating four respective terminal variables.

The first, on the right side of the micro level, is 'selective incentives to members of group (i.e., side payments) can increase contributions to provision of collective good'; 'smaller group size makes members more likely to contribute to provision of collective good'; 'group is 'privileged' if at least one member is willing to take on a disproportionate burden (i.e., exploitation of the great by the small) to ensure provision of collective good' → 'provision of collective good is improved or even optimal in some instances'. This is the positive outcome to be expected when one or more conditions are in place to overcome the problem of free riding.

The second, at the center of the micro level, is 'greater incentives toward free riding' → 'suboptimal provision of public good as a result of free riding—the logic of collective action'. As per the above, in the absence of ameliorating conditions, the free-rider effect is anticipated to be pervasive.

The third, at the macro level, is 'SUBOPTIMAL PROVISION OF PUBLIC GOOD AS A RESULT OF FREE RIDING—THE LOGIC OF COLLECTIVE ACTION' → 'STATE CANNOT CONTINUE TO EXIST AND PROVIDE COLLECTIVE GOODS WITHOUT TAXATION'. Consider the emerging states of early modern Europe. Improvements in artillery enabled would-be monarchies to defeat walled towns and other entities (McNeill 1982). As a result, modern states emerged because of an ability to collect taxes from those who otherwise would not be inclined to pay.

The fourth, and last among the pathways, is one that reaches from the macro level of the state into the international system: 'SUBOPTIMAL PROVISION OF PUBLIC GOOD AS A RESULT OF FREE RIDING—THE LOGIC OF COLLECTIVE ACTION' → 'COLLECTIVE ACTION PROBLEM VIA FREE RIDING EXISTS AT AN INTERNATIONAL

LEVEL AS WELL (E.G., SUBOPTIMAL BUDGET ALLOCATIONS AMONG NATO MEMBER STATES)'. The logic of collective action applies to states as well as individual people. Research on NATO, for example, produced results that supported free riding (Olson and Zeckhauser 1966), and a wide range of extensions and critiques in the decades that followed (e.g., Boyer 1993).

## 4. Contributions to the Special Issue

Six studies that represent connection points from IRs within the social sciences will appear in this Special Issue, followed by a concluding essay that sums up what has been accomplished in a collective sense. As a discipline, IR is ideal as a starting point for the application of the systemist graphic method throughout the social sciences. This is because IRs, for a very long time, have been recognized as a hybrid field of study (Wright 1955). Starting with Law and Philosophy, and moving on to Economics and Psychology, points of contact for IRs have expanded over the decades to include a wide range of disciplines—even those of the natural sciences (Yetiv and James 2016).[10] The studies in this Special Issue represent connections with, respectively, the following fields and interdisciplinary areas of research: education (Parmentier),[11] organizations (Genna), demography and geography (Kugler and Rhamey), foreign-policy analysis (Redd), psychology (Ben-Nun Bloom and Gregor), and religion (Akbaba). Each of the preceding contributions, along with the concluding essay by Gansen and James, is summarized in turn.

Parmentier observes that de-colonizing the curriculum and active learning approaches that engage students experientially are both current themes in the teaching of International Studies and related disciplines. For the discipline of Global Development, both are critically needed approaches to train students who are able to work across national contexts and effectively interact with communities of different political histories and cultures. Yet, neither is necessarily straightforward. Parmentier explores two pedagogical projects that, while very different from each other, reveal commonalities with a technique of systemist notation and visualization, strengthening the contribution to cross-cultural and cross-national active learning. While online international collaborations and study-abroad programs are different pedagogical contexts, they both involve significant levels of intercultural communication and knowledge exchange, neither of which is a given and requires careful course design and implementation.

Genna compares two articles that seek to explain why states participate in regional integration organizations and why they want to deepen their economic and political partnerships. The method of comparison is the systemist diagrammatic approach, which requires the deconstruction and mapping of ideas found in social science. The articles demonstrate common variables in their explanations, namely, that power asymmetry and satisfaction with the status quo among regional partners are critical to determining why states integrate. The articles diverge in their explanations: one emphasizes the similarities of institutionalized policies, and the other, the role that a rising power, China, has in developing regional integration in the Western Hemisphere.

Kugler and Rhamey assert that the disciplines of political demography and geography examine the interplay among social behaviors, spatial dimensions, politics, and policy. Investigations into demographic shifts, driven by evolving social norms or domestic and international political events, can influence numerous critical dependent variables in international relations, such as trade, development, and inter- and intra-state conflict. Similarly, geography and the interconnection of space with independent variables like power, wealth, and culture yield similar insights. Kugler and Rhamey employ a systemist approach to provide a brief overview of the theoretical intersection among Geography, Gemography, and IR. Systemist graphics of these seminal articles demonstrate the clear applicability of demography and geography to international politics scholarship.

The poliheuristic theory of foreign-policy decision making, as Redd points out, incorporates conditions surrounding foreign-policy decisions, as well as the cognitive processes decision makers undergo en route to a choice. Redd argues that high-level decision makers,

who routinely face stressful decision environments, engage in a two-stage decision process wherein they first employ cognitive-based, heuristic shortcuts in an attempt to simplify the decision task. In the second stage, once the decision task is more manageable, decision makers employ more analytic strategies in order to minimize risks and maximize rewards. Poliheuristic theory also posits that politics is the essence of decision and that decision makers avoid choosing alternatives that hurt them politically. Using systemist graphics, Redd compares two journal articles that use poliheuristic theory to explain foreign-policy behavior and choices. Systemism thereby enables Redd to precisely examine how poliheuristic theory has evolved over nearly twenty years and compare what the different methodologies of case studies and experimental methods have to offer in explaining the foreign-policy behavior of leaders and their advisers.

By conceptualizing it as a ground for moral threat, Ben-Nun Bloom and Gregor present a theoretical framework for understanding the social consequences of polarization-driven behavior. Their argument is presented using systemist graphics, which illustrate key connections and patterns from two distinct works of the scientific literature. A systematic synthesis of the two articles uncovers a twofold role of morality in polarization: as a factor in forming partisan animosity and a catalyst in maintaining and deepening it. Ben-Nun Bloom and Goldner further highlight the role of outgroup hate, rather than ingroup love, in driving negative actions resulting from polarization, and the difficulty of reconciling conflicts driven by moralization. The framework in graphic form sheds new light on the complex interplay between morality and conflicts, with implications for social cohesion, erosion of moral values, and democratic backsliding.

One of the established trends in the scholarship on religion, as Akbaba observes, is awareness of a rising level of religious discrimination against minorities. Although there is variation in rates, religious restrictions are widely observed, including in Western democracies. Scholarship about restrictions on the exercise of religion advanced by seeking answers to the following questions: Who discriminates? Who is discriminated against more? What are the causes of restrictions on religious freedom? What are the forms of discrimination? Akbaba answers those questions by connecting theories and findings of two religious discrimination studies of IR with the graphic method of systemism. Featured works engage with religious discrimination in a sub-group of states—Western democracies and Christian majority. While one study focuses on government-based restrictions, the other one engages with societal ones. Collectively, these works reveal the fragility of freedom and the importance of understanding the mechanisms that protect it.

Gansen and James assess what has been accomplished, in a collective sense, by the preceding set of contributions. Their concluding essay also offers ideas about future application of systemism to the social sciences.

## 5. Summing up and Moving Forward

Systemism is a perspective with much to offer to the social sciences vis-à-vis improved communication with a graphic approach. The VIRP provides an archive of graphics for one discipline already and that could be replicated for others. A classic work, *The Logic of Collective Action* (Olson 1965), has been used to demonstrate the systemist graphic approach in action. In the pages that follow, contributors will apply systemist methods to learn more about a range of subject areas.

**Author Contributions:** Each author contributed equally to all aspects of the article. All authors have read and agreed to the published version of the manuscript.

**Funding:** This research received no external funding.

**Institutional Review Board Statement:** Not relevant.

**Informed Consent Statement:** Not relevant.



**Data Availability Statement:** The graphics used in this article are the data and are fully available for anyone to access.

**Conflicts of Interest:** The authors declare no conflict of interest.

## Notes

1   Bunge (1996) made use of rudimentary 'box and arrow' diagrams to illustrate the systemist commitment to developing explanations that spanned levels of analysis for a given academic discipline. Detailed introductions to systemist graphics in IRs, which include a comprehensive set of rules for the construction of commensurable diagrams, appear in James (2019a, 2019b, 2022, 2023), Pfonner and James (2020), and Gansen and James (2022a, 2022b).

2   Research in educational psychology (e.g., Pashler et al. 2009) confirms that an explanation offered in both words and visual form can enhance learning and retention of knowledge, with the specific balance that is optimal varying from one discipline to the next. For a more detailed review of the literature in educational psychology, see James (2022).

3   The diagrammatic exposition that follows is primarily based upon James (2019a).

4   The term 'variable' is used throughout the exposition as a matter of convenience. It is understood that scholars who do not identify as empiricists would not see their work as being properly represented by a network of variables. Systemist visualizations of their scholarship should be regarded as depicting components of an argument as it unfolds rather than a set of equations.

5   Beyond the scope of the present exposition is the specification of the functional form for the proposed connections; this is required by systemism to completely articulate a theory (Bunge 1996). While incremental change is assumed as the default position, it is important to recognize that functional relationships can also be non-linear.

6   An established presence in the social sciences, social constructivism developed from its origins in the symbolic interactionism of Goffman (1956) in Sociology.

7   The connection of systemist graphics and causation is a topic beyond the scope of the present study. Such matters are taken up at length in James (2022).

8   If an author is deceased, former students can be consulted on the content and meaning of publications; if that option is not available, experts on the work of a given scholar can be contacted instead. For more detailed expositions on the creation of systemist figures, see Gansen and James (2022a) and James (2023).

9   A point of entry into the enormous literature stimulated by Olson (1965) is provided by the *festschrift* edited by Heckelman and Coates (2010).

10   Yetiv and James (2016) also explored the intersection of IRs and International Studies, and other related and overlapping fields.

11   Throughout this introductory essay, contributions to the Special Issue will be cited with name of author(s) in parentheses.

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
