# Peer review of "Introduction: Special Issue on the Visual International Relations Project"

_socsci, doi:10.3390/socsci12090498_

Round 1

Reviewer 1 Report

The paper is an excellent introduction to the special issue. It starts strong and is quite convincing. The second paragraph makes me want to keep reading: “Items in the VIRP archive depict books, articles and book chapters that connect with a wide range of disciplines in the social sciences and even beyond into the humanities and social sciences”. The issue seems perfect for a broad social science graduate course, and as mentioned, students preparing for exams. The choice of the Logic of Collective Action is brilliant, and its application seems flawless. The river analogy is another great move that works perfectly.

I have only three concerns. I know the authors have thought about these issues far more than I have, so I raise them very tentatively. I expect the authors to be the better judge on if, and how, they may be addressed.

The first is footnote 4. What is the difference of “an argument as it unfolds” versus “a configuration of variables that represent cause and effect”? I understand the authors are trying to reach a broad audience here, and this footnote might be the best that can be achieved. But I do not understand it.

The second is also in footnote 4: the use of the word ‘positivist’. I believe a ‘positivist’ is someone who believes that causation can be seen; a ‘post-positivist’ believes causation cannot be seen. I believe all published empirical research, at least in my field of IR, is in the post-positivist tradition, with scholars sharing what they believe to be objective facts from which causation can be tentatively inferred (I am following Vasquez on this one; Popper would be post-positive). I think the word ‘positivist’ is sometimes meant as a pejorative, to suggest that anyone who does empirical research thinks they can see causation. I could be wrong on this one, but the authors might consider the issue. Perhaps replace the word ‘positivists’ with ‘empiricists’?

Lastly, the authors might consider explaining a bit more about necessary and sufficient conditions and permissive and effective causes (line 142-153). Why would “just one pathway” imply “all conditions are deemed necessary”? If a theory predicts X causes Y, it does not imply that only X can cause Y (that X is necessary for Y). Why would all variables in the final pathways toward terminal variables be permissive? What is an ‘effective’ cause? I recognize the difficulty here in addressing the broader social sciences where terms are used quite differently across the fields. Can this be explained a few more sentences? Or maybe a 2X2 table?

I love the qualification at line 345. This is a great introduction to what will be an awesome special issue.  

Author Response

We are very grateful to the referee for a very positive review, along with drawing attention to the respective areas of improvement.  Each of the suggestions has been implemented and is visible through track changes.

One point of explanation is in order about the response to the referee’s observation about the issue of cause and effect (i.e., lines 142-153 of the initial draft).  It would require too much space to respond to the queries posed here, so the material in question has been deleted.  Instead, note 7 refers to the extended treatment in James (2022) of cause and effect in connection with systemist diagrams.

Reviewer 2 Report

Dear authors, 

this paper presents an intriguing approach, systemism, which has the potential to transform communication and understanding within the social sciences. By emphasizing graphical representations and a comprehensive view of causal relationships, the authors demonstrate how this approach can facilitate engagement, collaboration, and growth within the field. The inclusion of practical examples, such as the application to Olson's work, adds depth and clarity to the presented concepts. The paper's structure is well-organized, leading the reader through the introduction, core principles of systemism, applications, and a detailed example, culminating in a cohesive and insightful exploration of the subject.

My only suggestion is to improve figures, as the are hardly legible.

Best regards,

Reviewer

Author Response

We thank the referee for this very positive review.  The question of legibility with regard to the figures is being taken up with the editor of the journal.